# Una Visión Unificada de Transformaciones Biyectivas en la Optimización de Problemas de Permutaciones

**Mikel Malagón**[1], **Aimar Barrena**[1], **Hugo F. Iñigo**[1], **Josu Ceberio**[1], **Ekhiñe Irurozki**[3], and **Jose A. Lozano**[1,2]

[1]University of the Basque Country UPV/EHU, {mikel.malagon, hugofederico.inigo, josu.ceberio, ja.lozano}@ehu.eus, abarrena032@ikasle.ehu.eus
[2]Basque Center for Applied Mathematics (BCAM), jlozano@bcamath.org
[3]Telecom Paris, irurozki@telecom-paris.fr

## Abstract

Muchos problemas de optimización utilizan permutaciones para representar soluciones. Sin embargo, a pesar de su aparente simplicidad, las permutaciones presentan desafíos significativos para los métodos de optimización, especialmente para los algoritmos *Global Random Search (GRS)*. En particular, la restricción de exclusividad mutua presente en las permutaciones plantea un reto importante tanto para el aprendizaje como para el muestreo de distribuciones de probabilidad.

Un enfoque alternativo implica el uso de funciones biyectivas sobre el espacio de permutaciones ($\mathbb{S}_n$) que permite transformar las soluciones codificadas como permutaciones en representaciones de vectores enteros, conocidas como *inversion vectors*. Aunque los *inversion vectors* existen desde hace décadas, carecen de un marco general que proporcione una notación unificada y completa. En este artículo, presentamos una notación unificada que permite recodificar todos los posibles *inversion vectors*, y que sienta las bases para su caracterización. Por último, estudiamos el comportamiento de los algoritmos *GRS* cuando se utilizan diferentes *inversion vectors* en diversos problemas de permutaciones.

## 1 Introducción

Las permutaciones son vectores de elementos dentro de un conjunto finito, donde cada elemento aparece exactamente una vez. Debido a su versatilidad, las permutaciones son fundamentales en muchos problemas de optimización del mundo real [1], ya que se utilizan para representar rutas, órdenes, clasificaciones, emparejamientos y asignaciones. El *Traveling Salesman Person Problem (TSP)*, el *Quadratic Assignment Problem (QAP)*, el *Flowshop Scheduling Problem (PFSP)* y el *Linear Ordering Problem (LOP)*, son algunos de los problemas más representativos.

A pesar de su sencillez, las permutaciones se consideran una representación particularmente compleja de manejar para muchos algoritmos de optimización debido a varios factores: (1) son objetos inherentemente restringidos, ya que requieren que cada elemento del conjunto codificado aparezca exactamente una vez (la restricción de exclusividad mutua), (2) cualquier dependencia entre los elementos puede permanecer oculta debido a esta restricción, y (3) el número de permutaciones posibles crece exponencialmente con su longitud.

Los problemas de permutaciones representan un desafío importante para los algoritmos *Global Random Search* (GRS) [2]. Estos métodos se basan en un muestreo secuencial de distribuciones de probabilidad, donde las soluciones de alta calidad extraídas de $P_t$ en la iteración $t$ se utilizan para construir la distribución $P_{t+1}$ de la siguiente iteración. La construcción de $P_t$, y particularmente su capacidad para muestrear soluciones relevantes para el problema, es un componente crítico que influye directamente en el comportamiento de los algoritmos GRS.

XVI XVI Congreso Español de Metaheurísticas, Algoritmos Evolutivos y Bioinspirados (maeb 2025).

Algunos de los paradigmas de optimización más conocidos tales como la optimización Bayesiana, *model-based gradient search*, y los algoritmos evolutivos como *ant colony optimization*, *particle swarm optimization* y los Algoritmos de Estimación de Distribuciones (EDAs) [3], pertenecen a la clase GRS de algoritmos.

Como se mencionó anteriormente, para que los algoritmos GRS optimicen eficazmente problemas de permutaciones, es deseable que la base de las distribuciones $P_t$ sea el grupo simétrico $\mathbb{S}_n$. Cabe destacar que no todos los modelos probabilísticos que definen una distribución sobre $\mathbb{S}_n$ son igualmente adecuados para cada problema de permutaciones. Por ejemplo, aunque tanto el TSP como el QAP representan soluciones como permutaciones, la naturaleza distinta de sus soluciones —rutas en el TSP y asignaciones en el QAP— requiere diferentes modelos probabilísticos [4].

Por este motivo, se han adoptado modelos específicos para el dominio de las permutaciones como el modelo de Mallows (y sus versiones generalizadas)[5], Plackett-Luce [6], Bradley-Terry [7] o las matrices doblemente estocásticas [4]. Todos ellos han demostrado rendimientos notables sobre los problemas donde se han aplicado. Sin embargo, todos comparten una limitación común: implican procesos numéricos costosos en las etapas de aprendizaje o muestreo.

En una línea de trabajo paralela, algunos estudios plantean utilizar funciones biyectivas en $\mathbb{S}_n$ para transformar soluciones codificadas como permutaciones en representaciones alternativas de vectores de números enteros, donde la restricción de exclusividad mutua no se aplica explícitamente [8]. Estas representaciones, conocidas como *inversion vectors*, codifican una permutación de longitud $n$ como un vector entero de la misma longitud, donde el $i$-ésimo elemento del vector indica la cantidad de elementos mayores (o menores) que el $i$-ésimo elemento de la permutación (o su inversa) ubicados a su derecha (o izquierda).

El concepto de *inversion vectors* se remonta a los años 1800 [9] y ha sido aplicado en numerosos trabajos de optimización combinatoria [8]. Sin embargo, la literatura muestra una notable falta de consistencia en las definiciones y notaciones para los diferentes tipos de *inversion vectors*. Diversos estudios se refieren a los mismos tipos de *inversion vectors* utilizando terminologías y formalismos distintos ([10] y [11], o [12] y [13]). Por último, se ha observado que el uso de *inversion vectors* afecta al rendimiento de los algoritmos de optimización, en ocasiones positivamente, en comparación con el uso directo de permutaciones en ciertos problemas [14, 15].

En este artículo, presentamos las siguientes contribuciones. En primer lugar, proporcionamos una perspectiva unificada de todas las posibles codificaciones de *inversion vectors* para permutaciones, definiendo formalmente todos sus tipos e introduciendo una notación coherente que resuelve las inconsistencias existentes en la literatura. En segundo lugar, establecemos transformaciones biyectivas entre diferentes *inversion vectors*, lo que facilita la categorización y el análisis de sus relaciones y características. Por último, demostramos experimentalmente que el rendimiento del algoritmo UMDA [16] (un EDA clásico) varía entre problemas según el *inversion vector* utilizado. Si bien existen trabajos previos similares [14, 15], en este artículo relacionamos el rendimiento en optimización con una categorización teórica de las transformaciones biyectivas que presentamos en este trabajo.

## 2   Preliminares

Una permutación de un conjunto finito es un vector que contiene todos los elementos del conjunto dispuestos en un orden determinado. Por ejemplo, dado el conjunto $S = \{A, B, C, D\}$, una posible permutación es $\sigma = (B, C, D, A)$ o $\pi = (C, D, B, A)$. Siguiendo la convención común en la literatura, las permutaciones se denotan con letras griegas minúsculas. Según esta definición, una permutación debe contener todos los elementos del conjunto, sin repetir ninguno, lo que se conoce como la *restricción de exclusividad mutua*.

Formalmente, una permutación es una biyección de un conjunto $S$ en sí mismo, $\sigma : S \to S$. El conjunto de todas las permutaciones de longitud $n$ forma el grupo simétrico, denotado como $\mathbb{S}_n$, y su cardinalidad es $n!$. En el resto del artículo, consideramos permutaciones del conjunto $1, 2, \ldots, n$, donde $n$ es la longitud de las permutaciones. En este caso, el primer elemento de una permutación es $\sigma(1)$, y el último es $\sigma(n)$. Así, la permutación identidad de longitud $n$ se define como $\sigma(i) = i$.

Una permutación $\sigma$ puede componerse con otra permutación $\pi$, denotándose como $\tau = \sigma \circ \pi$, y definida por $\tau(i) = \sigma(\pi(i))$. Es importante destacar que la composición no es conmutativa, es decir,

$\sigma \circ \pi \neq \pi \circ \sigma$, y que la composición de cualquier permutación con la identidad $e$ da como resultado la misma permutación: $\sigma = \sigma \circ e = e \circ \sigma$.

Alternativamente, la inversa de una permutación $\sigma$, denotada como $\sigma^{-1}$, satisface la propiedad $\sigma(i) = j \Leftrightarrow \sigma^{-1}(j) = i$ para todo $i, j \in 1, \ldots, n$. Por ejemplo, dada $\sigma = (3, 1, 2, 4)$, su permutación inversa es $\sigma^{-1} = (2, 3, 1, 4)$. Finalmente, la composición de una permutación con su inversa siempre da como resultado la permutación identidad: $(\sigma \circ \sigma^{-1} = \sigma^{-1} \circ \sigma = e)$.

## 3   *Inversion Vectors*: **Notación unificada**

Los *inversion vectors* se remontan a [9]. Desde entonces, estas estructuras combinatorias han sido definidas y referidas de diversas maneras. Por ejemplo, [17] y [11] se refieren a este tipo de *inversion vector* como la *tabla de inversión* y el *vector de inversión*, respectivamente. Sin embargo, [18] usa los términos *vector de inversión* o *tabla de inversión* para un tipo diferente de *inversion vector*, que [19] denomina *right inversion vector*. Además, lo que [19] llama *left inversion vector* es denominado *código Lehmer* por [12].

Por otro lado, los trabajos de [20, 21] y [22] sobre modelos probabilísticos de permutaciones se refieren a la misma estructura como "un modelo de elección de componentes independientes" y lo vinculan con la distancia Kendall-$\tau$ en permutaciones. De manera similar, [8] presentó el *repetead insertion model*, apuntando también a una transformación biyectiva de $\mathbb{S}_n$ como "un modelo de elección", pero sin mencionar las referencias y términos previos, algunos de los cuales datan de años atrás.

En respuesta al actual desacuerdo sobre la notación y los formalismos relacionados con los *inversion vectors* en la literatura, presentamos a continuación una notación clara para unificar todos los tipos de "transformaciones biyectivas de $\mathbb{S}_n$" y nos referiremos a ellas con un único término: *inversion vectors*. La definición y notación propuestas buscan proporcionar una visión unificada y servir como base para las contribuciones en las secciones siguientes.

Aunque existen variaciones en las definiciones de *inversion vectors*, todas coinciden en la siguiente descripción: dada una permutación $\sigma$, el $i$-ésimo elemento de cualquier tipo de *inversion vector* cuenta el número de elementos en $\sigma$ (o $\sigma^{-1}$) que son mayores (o menores) que $\sigma(i)$ (o $\sigma^{-1}(i)$) a su derecha (o izquierda).

Por ejemplo, los *inversion vectors* de [19] y [12] son aquellos cuyo $i$-ésimo elemento cuenta el número de elementos en $\sigma$ mayores que $\sigma(i)$ a su izquierda y se expresan matemáticamente como

$$\overleftarrow{V}_{\sigma}^{>}(j) = \sum_{i=1}^{j} \mathbb{I}[\sigma(i) > \sigma(j)], \tag{1}$$

donde $\mathbb{I}$ es la función indicadora[1]. Siguiendo este enfoque, dada la permutación $\sigma = (2, 3, 1, 4)$, el vector de inversión correspondiente es $(0, 0, 2, 0)$. Denotamos este vector de inversión específico como $\overleftarrow{V}_{\sigma}^{>}$, para señalar que cada elemento del vector cuenta los elementos mayores (denotado por el superíndice $^{>}$) que $\sigma(i)$ en $\sigma$ (denotado por el subíndice $\sigma$) a su izquierda (indicado por la flecha $\leftarrow$ sobre $V$). De manera similar, un vector de inversión que considera los elementos menores a la derecha de $\sigma$ (como los de [9], [18], [19]) se denota como $\overrightarrow{V}_{\sigma}^{<}$.

El *inversion vector* de una permutación $\sigma$ también puede considerar las relaciones entre los elementos en $\sigma^{-1}$ (ver [17, 11, 20, 21, 22]). Por ejemplo, la Ec. 1 redefinida para considerar los elementos mayores en $\sigma^{-1}$, en lugar de $\sigma$, sería referida como $\overleftarrow{V}_{\sigma^{-1}}^{>}$ y definida formalmente como

$$\overleftarrow{V}_{\sigma^{-1}}^{>}(j) = \sum_{i=1}^{j} \mathbb{I}[\sigma^{-1}(i) > \sigma^{-1}(j)] \tag{2}$$

En este caso, dada la permutación $\sigma = (4, 1, 3, 2)$, su permutación inversa es $\sigma^{-1} = (2, 4, 3, 1)$, y $\overleftarrow{V}_{\sigma^{-1}}^{>} = (0, 0, 1, 3)$.

---

[1]Cuando la condición interna se cumple, el valor de la función es 1, de lo contrario es 0.

Tabla 1: Transformaciones en forma cerrada entre todos los posibles *inversion vectors* definidos en la Sección 3. Cada celda $(j,k)$ de la tabla presenta la ecuación que relaciona elemento por elemento el *inversion vector* de su columna $k$ con el de su fila $j$. Las celdas marcadas con '?' indican casos donde las transformaciones biyectivas aún no han sido descubiertas.

| To \ From | $\overleftarrow{V}^<_{\sigma^{-1}}$ | $\overleftarrow{V}^>_{\sigma}$ | $\overleftarrow{V}^>_{\sigma^{-1}}$ | $\overrightarrow{V}^<_{\sigma}$ | $\overrightarrow{V}^<_{\sigma^{-1}}$ | $\overrightarrow{V}^>_{\sigma}$ | $\overrightarrow{V}^>_{\sigma^{-1}}$ |
|---|---|---|---|---|---|---|---|
| $\overleftarrow{V}^<_{\sigma}$ | $\overleftarrow{V}^<_{\sigma^{-1}}(\sigma(i))$ | $i-1-\overleftarrow{V}^>_{\sigma}(i)$ | $\sigma(i)-\overleftarrow{V}^>_{\sigma^{-1}}(\sigma(i))$ | $\sigma(i)-\overrightarrow{V}^<_{\sigma}(i)$ | $i-1-\overrightarrow{V}^<_{\sigma^{-1}}(\sigma(i))$ | $\sigma(i)-n+\overrightarrow{V}^>_{\sigma}(i)+i$ | $\sigma(i)+\overrightarrow{V}^>_{\sigma^{-1}}(\sigma(i))-n+i$ |
| $\overleftarrow{V}^<_{\sigma^{-1}}$ | | ? | $i-1-\overleftarrow{V}^>_{\sigma^{-1}}(i)$ | $i-1-\overrightarrow{V}^<_{\sigma}(\sigma^{-1}(i))$ | $\sigma^{-1}(i)-\overrightarrow{V}^<_{\sigma^{-1}}(i)$ | ? | $\sigma^{-1}(i)-n+\overrightarrow{V}^>_{\sigma^{-1}}(i)+i$ |
| $\overleftarrow{V}^>_{\sigma}$ | | | $\overleftarrow{V}^>_{\sigma^{-1}}(\sigma(i))-\sigma(i)+i-1$ | $\overrightarrow{V}^<_{\sigma}(i)-\sigma(i)+i-1$ | $\overrightarrow{V}^<_{\sigma^{-1}}(\sigma^{-1}(i))$ | $n-1-\sigma(i)-\overrightarrow{V}^>_{\sigma}(i)$ | $n-\overrightarrow{V}^>_{\sigma^{-1}}(\sigma(i))-\sigma(i)-1$ |
| $\overleftarrow{V}^>_{\sigma^{-1}}$ | | | | $\overrightarrow{V}^<_{\sigma}(\sigma^{-1}(i))$ | $\overrightarrow{V}^<_{\sigma^{-1}}(i)-\sigma^{-1}(i)+i-1$ | ? | $n-1-\sigma^{-1}(i)-\overrightarrow{V}^>_{\sigma^{-1}}(i)$ |
| $\overrightarrow{V}^<_{\sigma}$ | | | | | $\overrightarrow{V}^<_{\sigma^{-1}}(\sigma(i))+\sigma(i)-i+1$ | $n-i-\overrightarrow{V}^>_{\sigma}(i)$ | $n-i-\overrightarrow{V}^>_{\sigma^{-1}}(\sigma(i))$ |
| $\overrightarrow{V}^<_{\sigma^{-1}}$ | | | | | | ? | $n-i-\overrightarrow{V}^>_{\sigma^{-1}}(i)$ |
| $\overrightarrow{V}^>_{\sigma}$ | | | | | | | $\overrightarrow{V}^>_{\sigma^{-1}}(\sigma(i))$ |

Cabe señalar que la notación del vector que hemos proporcionado está sujeta a la variación de tres elementos ($\sigma$ o $\sigma^{-1}$, $\leftarrow$ o $\rightarrow$, y $>$ o $<$). Por lo tanto, el número total de vectores posibles que se pueden implementar es ocho. Específicamente, todos los tipos de *inversion vectors* se denotan como: $\overleftarrow{V}^<_{\sigma}$, $\overleftarrow{V}^<_{\sigma^{-1}}$, $\overleftarrow{V}^>_{\sigma}$, $\overleftarrow{V}^>_{\sigma^{-1}}$, $\overrightarrow{V}^<_{\sigma}$, $\overrightarrow{V}^<_{\sigma^{-1}}$, $\overrightarrow{V}^>_{\sigma}$, y $\overrightarrow{V}^>_{\sigma^{-1}}$.

# 4 Biyecciones entre *Inversion Vectors*

Dadas las definiciones mencionadas, podría pensarse que algunos *inversion vectors* comparten propiedades análogas. Por ejemplo, uno podría suponer que "contar los elementos menores a la izquierda" sería equivalente a "contar los elementos mayores a la derecha". En consecuencia, el objetivo de esta sección es caracterizar teóricamente las posibles propiedades y relaciones entre todos los *inversion vectors* definidos en la Sección 3. Para ello, en la Tabla 1 mostramos las transformaciones de cada definición de *inversion vector* (columnas) hacia las demás (filas). Es importante señalar que solo se presenta la diagonal superior y que la diagonal de la matriz ha sido eliminada, ya que la transformación de un *inversion vector* hacia sí mismo está dada por la función identidad.

A continuación, identificamos cuatro categorías distintas de biyección de entre las ocho posibilidades (ver colores en las celdas de la Tabla 1):

Categoría $\alpha$ incluye las biyecciones entre pares de *inversion vectors* en los que la conversión de un espacio a otro ocurre sin hacer referencia explícita a la permutación subyacente $\sigma$ que codifican. Estos pares son $\{\overleftarrow{V}^<_{\sigma}, \overleftarrow{V}^>_{\sigma}\}$, $\{\overleftarrow{V}^<_{\sigma^{-1}}, \overleftarrow{V}^>_{\sigma^{-1}}\}$, $\{\overrightarrow{V}^<_{\sigma}, \overrightarrow{V}^>_{\sigma}\}$, y $\{\overrightarrow{V}^<_{\sigma^{-1}}, \overrightarrow{V}^>_{\sigma^{-1}}\}$, y están marcados en verde en la tabla. Es importante destacar que, en cada par, se encuentran biyecciones de *inversion vectors* con el signo de comparación opuesto.

Categoría $\beta$ incluye las biyecciones que requieren sumar o restar el $i$-ésimo elemento de la permutación subyacente que los *inversion vectors* codifican. En este caso, están agrupados en las ternas $\{\overleftarrow{V}^<_{\sigma}, \overrightarrow{V}^<_{\sigma}, \overrightarrow{V}^>_{\sigma}\}$, $\{\overleftarrow{V}^>_{\sigma}, \overrightarrow{V}^<_{\sigma}, \overrightarrow{V}^>_{\sigma}\}$, $\{\overleftarrow{V}^<_{\sigma^{-1}}, \overrightarrow{V}^<_{\sigma^{-1}}, \overrightarrow{V}^>_{\sigma^{-1}}\}$, y $\{\overleftarrow{V}^>_{\sigma^{-1}}, \overrightarrow{V}^<_{\sigma^{-1}}, \overrightarrow{V}^>_{\sigma^{-1}}\}$, marcados en azul en la tabla.

Categoría $\gamma$ agrupa las biyecciones a partir de las cuales se puede transformar un vector de una definición a otra solo reordenando los elementos de los vectores. Esta categoría está formada por los pares $\{\overleftarrow{V}^<_{\sigma}, \overleftarrow{V}^<_{\sigma^{-1}}\}$, $\{\overrightarrow{V}^<_{\sigma^{-1}}, \overrightarrow{V}^>_{\sigma^{-1}}\}$, $\{\overleftarrow{V}^>_{\sigma}, \overrightarrow{V}^<_{\sigma^{-1}}\}$, y $\{\overrightarrow{V}^<_{\sigma}, \overleftarrow{V}^>_{\sigma^{-1}}\}$, que están marcados en naranja en la tabla. Es importante señalar que los primeros dos pares están compuestos por los *inversion vectors* con el mismo signo y dirección (solo en el caso de elementos menores a la izquierda y mayores a la derecha), que codifican la permutación inversa. Los otros dos pares están formados por las transformaciones restantes, donde cada una se empareja con su transformación opuesta.

Este grupo es clave para resolver las biyecciones de la siguiente categoría, ya que estas pueden encontrarse mediante las ecuaciones que combinan los grupos $\alpha$ y $\beta$ con $\gamma$. Para este proceso,

también hemos utilizado las biyeciones que corresponderían a la diagonal inferior de la Tabla 1:

$$\overleftarrow{V}_{\sigma}^{<}(i) = \overleftarrow{V}_{\sigma^{-1}}^{<}(\sigma(i)) \longleftrightarrow \overleftarrow{V}_{\sigma^{-1}}^{<}(i) = \overleftarrow{V}_{\sigma}^{<}(\sigma^{-1}(i))$$

$$\overrightarrow{V}_{\sigma}^{>}(i) = \overrightarrow{V}_{\sigma^{-1}}^{>}(\sigma(i)) \longleftrightarrow \overrightarrow{V}_{\sigma^{-1}}^{>}(i) = \overrightarrow{V}_{\sigma}^{>}(\sigma^{-1}(i))$$

$$\overleftarrow{V}_{\sigma}^{>}(i) = \overrightarrow{V}_{\sigma^{-1}}^{<}(\sigma(i)) \longleftrightarrow \overrightarrow{V}_{\sigma^{-1}}^{<}(i) = \overleftarrow{V}_{\sigma}^{>}(\sigma^{-1}(i))$$

$$\overrightarrow{V}_{\sigma}^{<}(i) = \overleftarrow{V}_{\sigma^{-1}}^{>}(\sigma(i)) \longleftrightarrow \overleftarrow{V}_{\sigma^{-1}}^{>}(i) = \overrightarrow{V}_{\sigma}^{<}(\sigma^{-1}(i))$$

Categoría $\delta$ es similar a la anterior, ya que agrupa transformaciones que requieren reordenar el vector, aunque también abarca transformaciones que requieren simultáneamente reordenar y sumar o restar los elementos de la permutación subyacente. La categoría está formada por ocho biyecciones, marcadas en rojo en la tabla, una para cada transformación. Un subconjunto está compuesto por la transformación en cuestión y todas las transformaciones con la permutación inversa que codifican, excepto aquella con la misma dirección y orden; por lo tanto, todos los subconjuntos tienen una cardinalidad de 4. Por ejemplo, uno de esos subconjuntos es $\overleftarrow{V}_{\sigma}^{<}, \overleftarrow{V}_{\sigma^{-1}}^{>}, \overrightarrow{V}_{\sigma^{-1}}^{>}, \overrightarrow{V}_{\sigma^{-1}}^{<}$.

## 5   Impacto en el rendimiento de los EDAs

El objetivo de esta sección es evaluar el rendimiento de las ocho transformaciones descritas en la Sección 3 cuando se aplican para optimizar problemas de permutaciones en el contexto de algoritmos GRS. En particular, utilizaremos como ejemplo el *Univariate Marginal Distribution Algorithm* (UMDA)[16], un EDA clásico, y los problemas de permutaciones que se emplearán como banco de pruebas son el PFSP [23], QAP [24] y LOP [25].

Concretamente, se ha utilizado la misma configuración del algoritmo UMDA para probar las ocho transformaciones: la población se compone de 100 soluciones, donde la mitad de las mismas se truncan para estimar la distribución de probabilidad y, en cada iteración, se muestrean 100 soluciones nuevas. El criterio de parada se limita a 100 iteraciones[2]; y se realizan 30 repeticiones con diferentes semillas aleatorias para cada par instancia - *inversion vector*.

Se han seleccionado, sin experimentación previa, tres instancias de referencia para cada problema, con tamaños que varían entre 20 y 100[3].

Los resultados se presentan en la Figura 1 en dos formas: (1) gráficos de convergencia que muestran el mejor valor de la función objetivo encontrado por los ocho algoritmos en los tres problemas (filas) y los tres tamaños de instancias (columnas), y (2) la densidad del mejor valor objetivo normalizado alcanzado por cada algoritmo, agregada a través de las instancias de cada problema.

En los gráficos de convergencia, observamos que solo aparecen tres curvas debido a las superposiciones significativas en los resultados. Específicamente, hemos identificado tres grupos de resultados: A, B y C. El grupo A corresponde a los algoritmos implementados con $\overleftarrow{V}^{>}\sigma$ y $\overleftarrow{V}^{<}\sigma$. El grupo B está compuesto por $\overrightarrow{V}^{>}\sigma$ y $\overrightarrow{V}^{<}\sigma$, mientras que el grupo C incluye las cuatro biyecciones restantes, es decir, aquellas que utilizan $\sigma^{-1}$. Estos grupos de resultados también se reflejan en las gráficas de densidad.

Las superposiciones de los algoritmos en los grupos A y B son completamente coherentes con las transformaciones presentadas en la Tabla 1 (ver celdas en verde). Es importante destacar que las transformaciones $\overleftarrow{V}^{>}\sigma$ y $\overleftarrow{V}^{<}\sigma$ capturan la misma información. De manera similar, las transformaciones $\overrightarrow{V}^{>}\sigma$ y $\overrightarrow{V}^{<}\sigma$ también representan la misma información. Como resultado, cuando se aprenden probabilidades marginales de primer orden (como en el caso del UMDA), la probabilidad de una solución $\sigma$ es la misma bajo los *inversion vectors* del grupo A, y lo mismo ocurre con los *inversion vectors* del grupo B. Al considerar una muestra de permutaciones, la probabilidad marginal de primer

---

[2]Cabe destacar que el propósito de este experimento es analizar el comportamiento del EDA; por lo tanto, la optimización se limita a 100 iteraciones, donde se logra en la mayoría de las ocasiones la convergencia del algoritmo.

[3]Las instancias de referencia específicas son $tai20\_5\_8$, $tai50\_10\_8$, $tai100\_20\_8$ para el PFSP [26], $chr20a$, $tai45e01$, $sko100a$ para el QAP [27], y $N - be75eec$, $N - stabu70$, $N - usa79$ para el LOP [25].

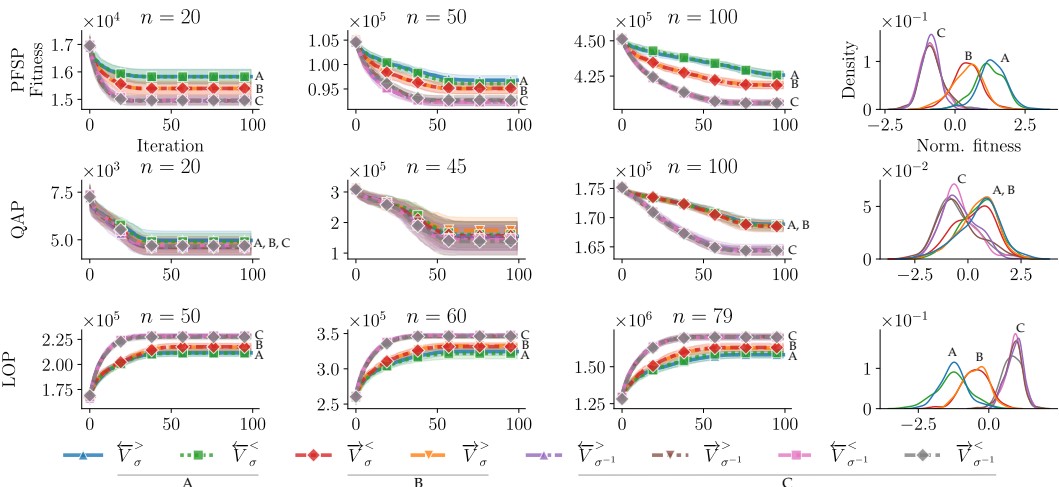

Figura 1: Comparación en términos del mejor valor de la función objetivo (*fitness*) encontrado por el UMDA para diferentes representaciones de *inversion vector*. Cada fila muestra los resultados obtenidos en el mismo problema de optimización combinatoria: de arriba a abajo, PFSP, QAP y LOP. Las tres primeras figuras (de izquierda a derecha) de cada fila muestran, para instancias de diferentes tamaños, el mejor *fitness* en cada iteración del UMDA sobre los distintos *inversion vectors*. En estos gráficos, las líneas representan la media y los contornos indican la desviación estándar de 30 repeticiones. La figura más a la derecha de cada fila presenta la densidad del mejor *fitness* normalizado alcanzado al usar diferentes *inversion vectors* en las instancias mostradas a su izquierda. Finalmente, en el caso del PFSP y QAP, un menor *fitness* es mejor, mientras que en el LOP, un mayor *fitness* es preferible.

orden de los *inversion vectors* $\overleftarrow{V}_\sigma^\leq$ y $\overleftarrow{V}_\sigma^\geq$ que codifican una permutación $\sigma$ es idéntica, derivándose de la biyección entre ellos. En este caso, se cumple que $P(\overleftarrow{V}_\sigma^\geq(i) = j) = P(\overleftarrow{V}_\sigma^\leq(i) = i - 1 - j)$.

El rendimiento de UMDA bajo los *inversion vectors* que consideran $\sigma^{-1}$ siempre es superior al obtenido con aquellos que consideran $\sigma$, lo cual resulta algo contra-intuitivo desde una perspectiva de optimización. Es importante resaltar que las tres funciones objetivo empleadas en este estudio consideran la solución de entrada $\sigma$ como una permutación de los objetos del problema (como fábricas, filas, trabajos, etc.). Sin embargo, la inversa de $\sigma$, $\sigma^{-1}$, reinterpreta la permutación como un ranking de dichos elementos. El cálculo del valor objetivo de una solución en su forma de ranking tiene una complejidad mayor en comparación con la interpretación directa de la permutación. Retomaremos esta idea en la siguiente sección para tratar de explicar este fenómeno.

En cuanto a las biyecciones que utilizan $\sigma$, es decir, los grupos A y B, observamos que el algoritmo obtiene mejores resultados con las que consideran los elementos a la derecha ($\overrightarrow{V}_\sigma^\leq$ y $\overrightarrow{V}_\sigma^\geq$) en lugar de aquellas que los consideran a la izquierda ($\overleftarrow{V}_\sigma^\leq$ y $\overleftarrow{V}_\sigma^\geq$), excepto en el caso del QAP, donde el rendimiento es el mismo en ambos casos.

## 6 Análisis de Resultados

Como se mencionó en la sección anterior, en algunos casos los resultados observados fueron consistentes con las definiciones proporcionadas. Sin embargo, el hecho de que los mejores resultados sean obtenidos por las transformaciones biyectivas que utilizan $\sigma^{-1}$ no parece tener sentido (grupo de resultados C). Esta sección se dedica a abordar esta cuestión. Para ello, a continuación realizamos una serie de estudios, que van desde la verificación de la corrección de los algoritmos hasta el análisis de por qué los *inversion vectors* que consideran $\sigma^{-1}$ capturan mejor las características que contribuyen a la calidad de las soluciones.

### 6.1 Estudio 1: Recuperación del modelo

El propósito de este estudio es doble. Por un lado, buscamos verificar la corrección de los métodos de aprendizaje y muestreo al recuperar un modelo probabilístico. Por otro lado, también pretendemos

analizar cuántas muestras son necesarias para recuperar el modelo probabilístico bajo cada transformación biyectiva. En ese sentido, hemos creado un gráfico para cada problema utilizando una instancia de $n = 50$ (ver Figura 2), para la que hemos generado una muestra de 150.000 permutaciones, seleccionando las 5.000 mejores según las funciones objetivo correspondientes en cada caso[4], y construido el modelo de probabilidad de referencia (matriz de frecuencias). Luego, realizamos una iteración de muestreo-aprendizaje con diferentes tamaños de muestra. En cada caso, calculamos la distancia de Frobenius entre el modelo de referencia y el generado en la iteración. Cada ejecución se llevó a cabo para las ocho transformaciones biyectivas estudiadas en este trabajo y se repitió cinco veces para evaluar la estocasticidad del experimento.

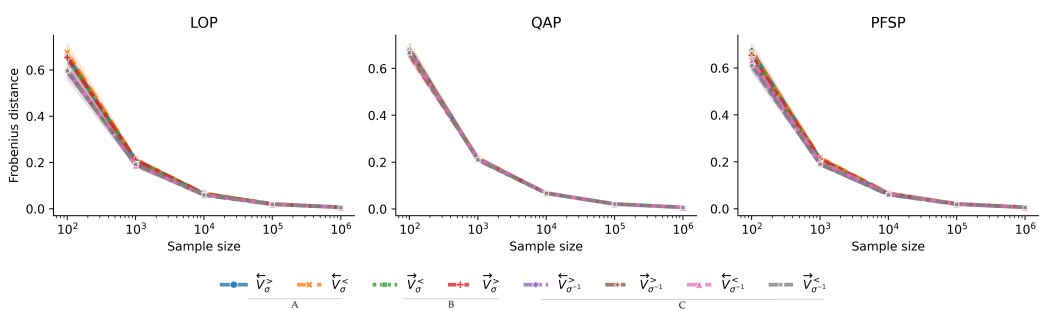

Figura 2: Gráficos de recuperación para una instancia aleatoria del LOP, QAP y PFSP. En cada gráfico, se han agregado cinco repeticiones del experimento con las ocho transformaciones.

Los resultados presentados en la Figura 2 muestran que (1) los métodos de aprendizaje y muestreo son correctos, ya que la diferencia tiende a cero conforme aumenta el tamaño de la muestra, y (2) todas las transformaciones recuperan los modelos de manera similar, es decir, no se observa una capacidad diferenciada entre las transformaciones para recuperar el modelo de referencia.

## 6.2 Estudio 2: Propagación de la calidad

Si un modelo es adecuado para optimizar un problema determinado, esperamos observar que la calidad de las soluciones muestreadas mejore a lo largo de las iteraciones. Para ello, para cada par instancia/modelo, trazamos los valores objetivos registrados durante una ejecución cada 10 iteraciones en un gráfico de densidad. Las curvas representan las diferentes iteraciones, de 0 a 50 (con un paso de 10 iteraciones). Para asegurar que la muestra inicial tuviera soluciones de "calidad suficientemente buena", y de manera similar al estudio anterior, generamos una muestra de 150.000 soluciones y seleccionamos las 5.000 mejores.

La última columna de la Figura 3 presenta los resultados observados en una instancia del LOP (los resultados para QAP y PFSP fueron similares y se han omitido debido a limitaciones de espacio).

En la Figura 3, es evidente que las transformaciones del grupo C lograron una mejora notable en los valores objetivos de las soluciones muestreadas a medida que avanzaban las iteraciones, mientras que dicha mejora es mucho menos evidente para los modelos de los grupos A y B. Considerando que los modelos iniciales en el experimento fueron aprendidos a partir de la misma selección de soluciones, la Figura 3 también muestra que las transformaciones del grupo C permiten una mejora más rápida y significativa de la calidad de estas muestras iniciales. En contraste, los modelos de los grupos A y B tienen más dificultades para mejorar las mismas muestras iniciales dentro del mismo presupuesto de iteraciones. De hecho, parece que el modesto progreso observado en las ejecuciones A y B proviene principalmente de la presión de selección, más que del modelo en sí.

En resumen, aunque las transformaciones del grupo $C$ parecen presentar patrones de probabilidad menos directamente alineados con la intuición inicial basada en $\sigma$ y tienen la correlación más baja con la función objetivo de los problemas (según nuestra interpretación expuesta en la Sección 5), resultan ser más efectivas que las demás.

---

[4]Las instancias para cada problema se han generado muestreando los parámetros correspondientes de manera uniforme.

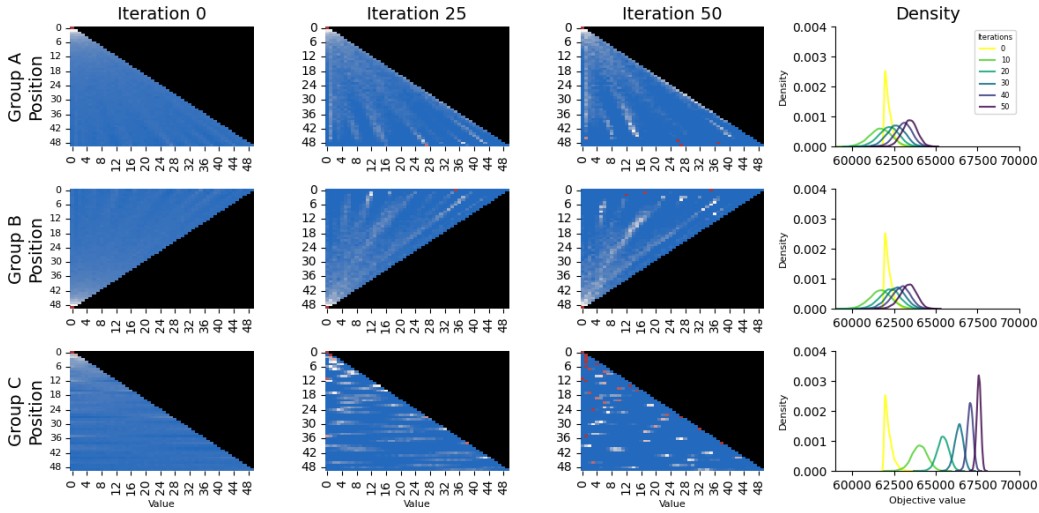

Figura 3: Las primeras tres columnas presentan los gráficos de *heat maps* de las probabilidades marginales en las iteraciones 0, 25 y 50. Los resultados de la primera fila corresponden a un grupo de resultados. Específicamente, las transformaciones $\overleftarrow{V}^{<}\sigma$, $\overrightarrow{V}^{<}\sigma$ y $\overleftarrow{V}^{<}_{\sigma^{-1}}$ han sido seleccionadas de los grupos de resultados A, B y C, respectivamente. La columna más a la derecha muestra los gráficos de densidad de los valores objetivos del LOP (instancia aleatoria de tamaño $n = 50$) registrados cada 10 iteraciones (empezando desde la iteración 0) utilizando las mismas biyecciones en cada grupo.

### 6.3  Estudio 3: Visualización de probabilidades marginales

Para tener una visión más detallada de los modelos generados en cada iteración y entender las razones por las cuales las transformaciones del grupo C tienen un mejor rendimiento, en la Figura 3, también presentamos las matrices de probabilidad en forma de *heat map*. Específicamente, y siguiendo la línea del estudio de ablación previo, nos hemos centrado en instancias aleatorias del LOP, y hemos visualizado los *heat map* de la matriz de probabilidades en las iteraciones 0, 25 y 50, para tres transformaciones diferentes, una por grupo: $\overleftarrow{V}^{<}\sigma$, $\overrightarrow{V}^{<}\sigma$, y $\overleftarrow{V}^{<}_{\sigma^{-1}}$.

Se observa que la entropía del modelo es mucho más baja para la transformación del grupo C, especialmente en las iteraciones 25 y 50. En estos casos, la probabilidad es mucho más dispersa para los grupos A y B. Esto se debe a que, en el caso del grupo C, el grado de convergencia del modelo es mayor, y es lógico pensar que las soluciones proporcionadas para el aprendizaje son más homogéneas. Este comportamiento es consistente con la menor varianza observada en los gráficos de densidad en el lado derecho del mismo gráfico.

Por otro lado, los *heat maps* generados para el grupo C presentan ciertos patrones horizontales que no se observan en los casos A y B. Al observar el mapa de calor de la iteración 50 para el grupo C, vemos que la probabilidad de los valores en cada posición se concentra en un conjunto de valores sucesivos en el rango $0, \ldots, n - 1$ (lo mismo ocurre en las otras iteraciones). Por ejemplo, la distribución de probabilidad para la posición 48 concentra la masa de probabilidad entre las posiciones 10 y 16. En cambio, en los casos A y B, se observan patrones diagonales, claramente influenciados por el hecho de que A y B utilizan $\sigma$, mientras que el caso C considera $\sigma^{-1}$ en la transformación.

## 7  Discusión

Los experimentos realizados revelaron que cuando se utilizan inversion vectors que emplean la versión $\sigma^{-1}$ de la solución, el algoritmo tiene un mejor rendimiento, converge más rápido y el modelo captura mejor la información que hace buenas las soluciones (en la mayoría de los casos observados, excepto en QAP, donde no hubo diferencias relevantes entre una opción y otra). Esto implica que, si usamos la representación $\sigma$, que es la que proporcionamos como entrada a las funciones objetivo, es conveniente reinterpretar la solución mediante la inversa, $\sigma^{-1}$, y luego aplicar la transformación.

La razón de este fenómeno puede estar en el siguiente hecho. Cuando usamos una transformación biyectiva del tipo descrito en este artículo, recodifica la permutación mediante la suma de ocurrencias de ciertos elementos a la izquierda (o a la derecha) que son mayores (o menores). De hecho, la recodificación requiere que los elementos que aparecen en la permutación tengan un carácter ordinal, ya que debemos hacer comparaciones de tipo mayor/menor.

Parece que esta propiedad es cierta tanto cuando usamos $\sigma$ como cuando usamos $\sigma^{-1}$; sin embargo, no es así, y esto puede deberse a la definición misma de la permutación que damos en la Sección 2. La representación $\sigma$ almacena en cada posición de la permutación las etiquetas de los trabajos, es decir, $\sigma(i) = j$, donde $j$ se refiere al trabajo $j$. Sin embargo, su inversa almacena la posición (clasificación) en la que los trabajos deben ser procesados, es decir, $\sigma^{-1}(i) = j$ se refiere a la posición en la que debe procesarse el trabajo $i$. Calcular la transformación biyectiva de la permutación $\sigma^{-1}$ es completamente factible, ya que un ranking es un conjunto de enteros (y por lo tanto operaciones tipo mayor/menos son posibles de llevar a cabo). Sin embargo, en el primer caso, donde la permutación $\sigma$ contiene las etiquetas de los trabajos, la aplicación de una transformación biyectiva puede carecer de sentido. Por ejemplo, supongamos que los trabajos están etiquetados con nombres de personas, "Juan", "Miguel" y "Manolo"; ¿qué sentido tendría calcular cuál es mayor (o menor) a la izquierda (o derecha)? Creemos que este es el motivo por el cual los algoritmos que emplean $\sigma$ en las transformaciones no funcionan correctamente.

## 8    Conclusión

Los *inversion vectors* han sido conocidos durante décadas y representan una valiosa alternativa a las permutaciones para codificar problemas de optimización combinatoria, ya que las transformaciones son biyectivas y no requieren cumplir con la restricción de exclusividad mutua que generalmente se impone en las codificaciones tradicionales. No obstante, muchos trabajos en la literatura han definido los *inversion vectors* (incluyendo este) de maneras diferentes, lo que ha generado confusión y desacuerdo sobre su definición, nomenclatura y uso.

Este artículo proporciona definiciones formales y una notación unificada para todos los tipos de codificación de *inversion vector*. Además, se presenta una tabla con las transformaciones biyectivas entre todos los tipos de *inversion vectors*, que se utiliza para caracterizar algunas de sus propiedades y los diferentes grupos que los conforman. A través de experimentos empíricos, mostramos que el rendimiento del UMDA [16] varía entre problemas dependiendo del *inversion vector* utilizado. No solo eso, sino que también verificamos que, por defecto, las transformaciones que utilizan $\sigma^{-1}$ ofrecen mejores resultados, probablemente debido a su idoneidad para ser modeladas en un proceso de optimización.

Aunque creemos que este artículo representa un avance significativo en la comprensión de los *inversion vectors* en el contexto de la optimización combinatoria, algunas preguntas clave aún permanecen abiertas. En particular, la Tabla 1 sigue incompleta y cuatro transformaciones aún no han sido resueltas. Finalmente, tanto el análisis teórico como empírico de este trabajo es preliminar. Creemos que investigaciones futuras en esta dirección proporcionarán conocimientos fundamentales sobre los problemas basados en permutaciones, lo que abrirá nuevas oportunidades para la mejora de algoritmos de optimización combinatoria.

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
