# OpenReview forum: "Una Visión Unificada de Transformaciones Biyectivas en la Optimización de Problemas de permutaciones"
_MAEB/2025/Congreso — MAEB 2025_

### Official Review · Reviewer_PPJ4 · 2025-03-11
**Revisión del Artículo: "Una Notación Unificada para Transformaciones Biyectivas en Problemas de Permutaciones"**

**Rating:** 5
**Confidence:** 4

**Review:**

El artículo propone una notación unificada para las transformaciones biyectivas en problemas de permutaciones, facilitando su análisis y categorización. Se demuestra que el uso de la permutación inversa mejora el rendimiento de los algoritmos de optimización, como el UMDA, en comparación con el uso directo de permutaciones. Los experimentos confirman que estas transformaciones permiten una mejor recuperación de modelos probabilísticos y una mayor calidad en las soluciones generadas. El artículo está bien escrito, recoge una revisión bibliográfica adecuada y presenta resultados experimentales sólidos.

El artículo debe ser aceptado si se corrigen ciertos aspectos menores, principalmente de forma o de redacción. A continuación, se detallan algunas recomendaciones para mejorar la calidad del artículo:

1. En las líneas 261-264 se menciona que las transformaciones del grupo C son significativamente mejores en términos de calidad comparadas con otros grupos. Sería conveniente respaldar esta afirmación con un análisis estadístico (por ejemplo, test de Wilcoxon, ANOVA) para confirmar la superioridad del grupo C desde un punto de vista estadístico.

2. El término "caótico" en la Sección 6.2 ("aunque las transformaciones del grupo C parecen ser las más caóticas...") puede llevar a una interpretación negativa o confusa. Sugiero matizar o redefinir "caótico" en este contexto. En lugar de "caóticas", podría ser más preciso describir las transformaciones del Grupo C como "menos intuitivamente ordenadas en su representación visual inicial" o "con patrones de probabilidad menos directamente alineados con la intuición inicial basada en σ".

3. Algunos errores ortográficos y de redacción menores:
- Los títulos de las secciones de introducción, discusión y conclusiones están en inglés.
- Línea 4: Global Random Search GRS --> Global Random Search (GRS).
- Línea 14: ahora podéis utilizar directamente el acrónimo GRS.
- Nota al pié de página 2 de la página 5: en la mayoria --> en la mayoría
- Línea 309: puede caracer de sentido --> puede carecer de sentido
- Línea 382: el nombre de la revista debería estar en el mismo formato que el resto de las referencias, en este caso European Journal of Operational Research. Recomiendo revisar la bibliografía, en general, porque no parece haber un formato consistente.

---

### Official Review · Reviewer_RLB5 · 2025-03-14
**Notación unificada para inversion vectors y su impacto en problemas de permutaciones**

**Rating:** 5
**Confidence:** 4

**Review:**

El artículo propone una notación unificada para recodificar los inversion vectors. La motivación principal es el uso de estos elementos en problemas relacionados con permutaciones. Es por ello, que para analizar la contribucción se estudian como afecta el uso de inversion vectors mediante los algoritmos Global Random Search en los problemas clásicos Linear Ordering Problem (LOP), Quadratic Assignment Problem (QAP) y Flow shop Scheduling Poblem (PFSP). La propuesta está bien escrita y considero que debería ser aceptado para el congreso. Abajo muestro algunos comentarios dónde considero que los autores deberían revisar, siendo los más relevantes el uso de inglés en lugares no adecuados y la necesidad de ajustarse al estilo/límite de referencias esperado para este congreso.

# Comentarios menores

- El título quizás abusa del uso de mayúsculas.
- (4, 15) Poner GRS como (GRS).
- (17,289,336) El artículo está en castellano, los títulos de las secciones deberían estar en castellano.
- (65) Separar: yPemmaraju = y Pemmaraju.
- (223) Cambiar Resultados por resultados.


# Comentarios

- (190) ¿Usar 3 instancias no son pocas? ¿Se hizo un estudio de características? ¿Son estructuralmente diferentes?
- (336) El formato de referencias no es el esperado, revisen la guía de estilo, tamaños máximo y modificadlo.

---

### Official Review · Reviewer_CkXE · 2025-03-17
**Propuesta de uso de inversion vectors para optimización de problemas con permutaciones**

**Rating:** 5
**Confidence:** 4

**Review:**

El trabajo presenta una notación unificada para inversion vectors, así como un análisis del comportamiento de los algoritmos GRS y, en concreto, el UMDA, sobre un conjunto de problemas clásicos de permutaciones para analizar el rendimiento de UMDA en esos problemas cuando se utilizan inversion vectors. El documento está correctamente redactado y presenta una descripción precisa y comprensible del problema tratado, acompañado de un análisis muy detallado de los resultados obtenidos. Además, realizan un estudio pormenorizado de las conclusiones obtenidas en cada experimento, resaltando en qué problemas es interesante el uso de estas estructuras y en cuáles no.

## Correcciones mayores:

En la sección 5 se indica que se utliza la misma configuración del algoritmo UMDA para las ocho transformaciones, pero ¿se podrían utilizar las mejores configuraciones de la literatura cuando se utilizan inversion vectors, o sería necesario ajustar de nuevo los parámetros por el cambio en el espacio de búsqueda? Se agradecería, si existe, una breve sección de recomendaciones en cuanto al ajuste de parámetros cuando se utilicen estas estructuras si ya se dispone de una configuración para la estructura original.

## Correcciones menores:

- l4: Global Random Search GRS -> Global Random Search (GRS)
- l12: permite recodifica -> permite recodificar
- l14: Global Random Search GRS -> GRS
- l18: En general, en castellano está más extendido el uso de "arrays" o "vectores" frente a "arreglos"
- l38: Uniformizar la notación: los nombres de los algoritmos se escriben con mayúsculas en algunas partes y en minúsculas en otras
- l89: Discussion -> cambiar el título a castellano

---

### Decision · Program_Chairs · 2025-03-19

Accept